# Alternative and Complementary Therapies against Foodborne Salmonella Infections

**DOI:** 10.3390/antibiotics10121453

**Published:** 2021-11-26

**Authors:** Mohamed F. Ghaly, Zahraa M. Nasr, Amira I. Abousaty, Hanan G. Seadawy, Mohamed A. A. Shaheen, Sarah Albogami, Mohammad M. Al-Sanea, Mahmoud M. Bendary

**Affiliations:** 1Microbiology Department, Faculty of Science, Zagazig University, Zagazig 44511, Egypt; Mfarouk2005@yahoo.com (M.F.G.); Zhraanasr96@gmail.com (Z.M.N.); Amira.abousaty83@gmail.com (A.I.A.); 2Agriculture Research Center (ARC), Animal Health Research Institute (AHRI), Zagazig 44511, Egypt; Hananseadawy70@gmail.com; 3Clinical Pathology Department, Faculty of Medicine, Al Azhar University, Chairo 11311, Egypt; dr.mashaheen85@gmail.com; 4Department of Biotechnology, College of Science, Taif University, Taif 21974, Saudi Arabia; dr.sarah@tu.edu.sa; 5Pharmaceutical Chemistry Department, College of Pharmacy, Jouf University, Sakaka 72341, Saudi Arabia; mmalsanea@ju.edu.sa; 6Microbiology and Immunology Department, Faculty of Pharmacy, Port Said University, Port Side 42511, Egypt

**Keywords:** *Salmonella*, foodborne, cinnamon, paprika, cefotaxime

## Abstract

The limitations in the therapeutic options for foodborne pathogens lead to treatments failure, especially for multidrug-resistant (MDR) *Salmonella* sp., worldwide. Therefore, we aimed to find alternative and complementary therapies against these resistant foodborne pathogens. Out of 100 meat products samples, the prevalence rate of salmonella was 6%, serotyped only as *S*. *Typhimurium* and *S. Enteritidis*. According to the antibiotic susceptibility assays, the majority of our isolates were MDR and susceptible to cefotaxime. Out of the 13 tested plant extracts, five only showed an inhibition zone in the range of 8–50 mm against both serotypes. Based on their promising activity, the oily extract of cinnamon and aqueous extract of paprika represented the highest potency. Surprisingly, a significant synergistic effect was detected between cinnamon oil and cefotaxime. Depending on Gas Chromatography/Mass Spectrometry (GC-MS), the antimicrobial activity of cinnamon oil was attributed to four components including linalool, camphor, (Z)-3-Phenylacrylaldehyde and its stereoisomer 2-Propenal-3-phenyl. The anti-virulence activities of these compounds were confirmed on the basis of computational molecular docking studies. Accordingly, we recommended the use of cinnamon oil as a food additive to fight the resistant foodborne pathogens. Additionally, we confirmed its therapeutic uses, especially when co-administrated with other antimicrobial agents.

## 1. Introduction

Meat and meat products are the most popular foods, and they provide an excellent source of human nutrition and a good source of high-class protein. On the contrary, they are the worst offenders when it comes to food poisoning, especially salmonellosis infection. *Salmonella* may also cause food poisoning, typhoid fever, gastroenteritis, enteric fever, and other illnesses [1]. Several antimicrobial preservatives are usually added to food to avoid infections with *Salmonella* sp. Unfortunately, there is evidence that these synthetic, preserved foods are also carcinogenic and toxic [2]. Additionally, there is a great incidence of *Salmonella* resistance to conventional antibiotics, which demands the addition of a high concentration of antibiotic preservatives on meat products to achieve the best biological activities [3,4].

Recently, the use of preservatives from natural sources has gained attention as an alternative to synthetic chemicals, due to their safety [5,6,7,8]. An unpleasant taste and other interactions of chemical preservatives prompted an increased interest in natural alternatives [9]. Aside from the lower toxicity and side effects of herbal extracts, they have strong antimicrobial activities. Therefore, there is a trend to replace chemical preservatives with other natural compounds [10]. Interestingly, the development of microbial resistance to plant extracts is more difficult than any commonly used chemical preservatives and antibiotics as the plant extracts have several antimicrobial components, which in turn have different target sites [11]. In the same context, it was announced that the antimicrobial activities of plant extracts were attributed to the leakage of essential cell components as a result of an increase in plasma membrane permeability and the inhibition of various cellular processes [12,13]. Of interest, essential oils are the safest and most effective natural food preservatives originating from plants. Therefore, essential oils may serve as an excellent antimicrobial agents due to several considerations: (i) they contain diverse groups of phytochemicals with multimodal actions; (ii) they penetrate microbial cells and cause alterations in their structure and function because of their hydrophobic nature; (iii) their functional diversity and diverse mechanisms of action hinder microbial resistance; (iv) some essential oils have strong antioxidant phenolic and γ-terpinene compounds [14]. Moreover, chemical preservatives are the main causes of hypersensitivity reactions when used in food and pharmaceutical preparation, even if they are added in small concentrations [15]. Increasingly, there is an urgent need for foods and pharmaceutical preparation to be free from chemical preservatives, especially for the allergic individuals. Therefore, essential oils can successfully be used instead of other chemical preservatives.

Importantly, the bioactive components of essential oils and other plant extracts must be identified to determine the best antimicrobial compounds. A bioautography assay is used to determine the active principles with a chromatogram. It combines a bioassay in situ with thin layer chromatography (TLC). It has many advantages such as a high efficiency for the separation of antimicrobial compounds [16].

Despite all the discussed therapeutic benefits of essential oils and other plant extracts, a high concentration of essential oils is required to achieve the desired in vivo antimicrobial activities [4]. Unfortunately, this concentration may cause negative organoleptic characteristics in meat, e.g., altering the texture, color, odor, and taste [17]. Successfully maintain the color of the meat is the major challenge for the use of these types of natural preservatives. We can overcome this problem via combination therapies, which have proven their potency in recent years and, especially, through the organoleptic impact of combination therapies, which allow essential oils to be incorporated in a wide range of food products. Additionally, new nanoencapsulated formulations may aid the wider applications of these compounds [18]. The cost/benefit ratios of using natural preservatives are considered in food industries. The high cost of natural preservatives can limit their use compared to synthetic preservatives. The increase in cost is the main issue with the complete replacement of synthetic preservatives by natural preservatives. Thus, the combination between different classes of natural preservatives, or with other chemical antibiotics, and the use of nanotechnology can reduce this issue [19].

The extraction methods are the critical point in the maintenance of the bioactive constituents from plant materials. Therefore, the antimicrobial activities of natural compounds are greatly affected by the extraction methods. Of note, there are no universal extraction methods that provide the maximum antimicrobial activates for the natural compounds. For each plant and target compounds, there is a unique extraction method. Essentially, the selection of the extraction method is very important and must be evaluated [20]. Therefore, our study was designed to measure the effectiveness of several herbal extractions alone or in combination with other antimicrobial agents, as well as the evaluation of the bioactive principles against multidrug-resistant (MDR) *Salmonella* sp. Additionally, the strengths and weaknesses of different extraction techniques were tested.

## 2. Results

### 2.1. The Prevalence Rate of Salmonella

Out of one hundred food samples, six isolates were identified as *Salmonella* via standard microbiological techniques and other genotypic methods. The serotyping of the *Salmonella* species revealed that only *S. Typhimurium* and *S. Enteritidis* (*N* = 3, each) were detected.

### 2.2. Initial Evaluation of Different Solvent Extracts

Regarding the aqueous extract, among the thirteen plants tested, five plants (paprika, cinnamon, thyme, bay leaf, and rosemary) recorded a significant antibacterial activity until the concentration reached 12.5% (Appendix A). The highest inhibition zone was detected with paprika. Meanwhile, the alcoholic extract of cinnamon showed the highest activity among the nine tested extracts against both *Salmonella* serotypes (Appendix A). Even among the seven tested oily extracts, cinnamon oil was also the only effective extract with an inhibition zone that reached 60 mm (Appendix A).

### 2.3. Antimicrobial Susceptibility

A disk diffusion assay revealed that all isolates were susceptible to cefotaxime, amoxicillin/clavulanic acid, and ciprofloxacin; a moderate susceptibility to gentamicin was also observed. A complete resistance to rifamycin was detected, and most isolates were multi-drug resistant (Figure 1). These results were confirmed depending on the MIC values (Appendix A). Cefotaxime was a highly effective antibiotic (MICs of 0.25–4 µg/mL) and MBC values ranged from (0.5–8 µg/mL) against the tested *Salmonella* isolates.

The comparative efficacy of all the tested extracts with antibiotics revealed that the oily extract of cinnamon and aqueous extract of paprika had highly significant antibacterial activities compared to cefotaxime. Additionally, the synergistic interactions among the most effective combinations of two plants and cefotaxime were evaluated in vitro, as shown in Figure 2 and Figure 3, where a remarkable synergistic effect was detected between cinnamon oil and cefotaxime.

### 2.4. Characterization of Cinnamon Oil Active Principal Compounds by Using TLC

The separation of compounds using TLC revealed the presence of different compounds in the cinnamon oil. The different compounds separated in TLC were collected by scraping the band from TLC, and the evaluation of their antibacterial potential was carried out against cinnamon oil. The promising compounds present in the Rf values (0.8 and 0.9) were further characterized (Appendix A and Figure 4).

According to the TLC-bioautography, ten bands were observed. One of the ten bands showed a maximum density, which was extracted for further investigation by Gas Chromatography/Mass Spectrometry (GC-MS).

### 2.5. Characterization of Compounds Present in the Oily Extract (Cinnamon Oil) by Using GC-MS

The characterization of compounds that are present in the effective extract revealed the presence of four antimicrobial compounds (Appendix A). The majority of compounds identified in the present study had an antimicrobial activity. Some of the compounds identified possessed anti-inflammatory, antidiabetic, antifungal, antioxidant, and antiprotozoal characteristics.

These are the chemical structures of the four bioactive components of cinnamon oil: linalool, camphor, (Z)-3-Phenylacrylaldehyde, and its stereoisomer 2-Propenal,3-phenyl, which were detected by Gas Chromatography/Mass Spectrometry (GC-MS).

### 2.6. Anti-Virulence Activities of Cinnamon Oil

Based on the obtained molecular docking results (Figure 5), N321 and R41 are highly significant for the interaction between small molecules and important salmonella virulence factors, such as cell invasion protein (SipD), since they are involved in both types of interactions (either through hydrogen bond formation (HB) or hydrophobic interactions). The co-crystallized ligand as well as Z-3-Phenylacrylaldehyde and its stereoisomer 2-Propenal,3-phenyl, (compound 6428995) demonstrated HB formation with an N321 amino acid residue, while camphor (compound 2537) showed a relatively strong hydrophobic interaction with it. Only linalool (compound 6549) showed potential HB with R41, while the other three studied compounds showed hydrophobic interactions. The following amino acid residues of salmonella’s SipD protein: I45, A108, and L322 were involved in hydrophobic interactions with at least three out of four compounds. Both the co-crystallized ligand and camphor showed hydrophobic interactions with V325 and K338, while the co-crystallized ligand and linalool had a similar hydrophobic interaction with N104. Camphor, as well as Phenylacrylaldehyde and its stereoisomer, 2-Propenal,3-phenyl, showed hydrophobic interactions with G42. The co-crystallized ligand showed a unique hydrophobic interaction with F340 and L318 amino acid residues, while linalool demonstrated a unique interaction with S107. The linalool also demonstrated a hydrogen bond formation with E39, while camphor, in its docked conformation, did not show hydrogen bonding formation with any of the amino acid residues of the SipD protein.

Based on both scoring functions of the ICM-PRO software, the co-crystallized ligand demonstrated relatively lower (better) scores than the studied ligands (Table 1). Phenylacrylaldehyde and its stereoisomer, 2-Propenal,3-phenyl, demonstrated the closest value to the co-crystallized ligand docking score from the three selected compounds.

## 3. Discussion

The infections with MDR bacteria [20,21,22] and fungi [23], especially foodborne resistant pathogens such as *Salmonella* typhimurium, *Salmonella enteritidis*, *Staphylococcus aureus*, *Campylobacter jejuni*, and *Listeria monocytogenes*, are considered as one of the most significant and severe health threats worldwide [24,25,26,27,28,29]. This is due to the high morbidity and mortality rates among the infected cases. The therapeutic options for these strains are limited, which results in the failure of treatments [30]. Nowadays, plant extracts and essential oils, in addition to drug repurposing, provide unlimited opportunities to manage these highly resistant pathogens [31,32,33].

The incidence of *Salmonella* sp. in this study was higher in contrast to other reports [34,35]. This difference might be attributed to differences in the hygienic and sanitary measures practiced in the respective abattoirs, especially municipal abattoirs. These types of abattoirs may have poor sanitation and hygienic standards in comparison with export abattoirs.

In recent decades, antimicrobial resistance (AMR) has become widely disseminated amongst food-borne *Salmonella* pathogens. Herein, a great resistance to rifamycin and sulfamethoxazole/trimethoprim was detected; this finding was most likely due to the careless use of antimicrobials in chicken feed or other environmental factors [36]. The therapeutic options for infections with the resistant pathogens, especially foodborne salmonellosis, are currently limited. Therefore, the surveillance of new strategies to manage this pathogen is an urgent need [37,38].

Nowadays, there is a renewed interest in the use of natural compounds in the treatment of MDR foodborne pathogens [8]. Numerous studies certified the antimicrobial activities of plant extracts. Of interest, almost all of the tested herbals in our research inhibited the growth of all *Salmonella* isolates and the most effective methodology of extraction was the use of essential oils, especially cinnamon oil. Several studies reported the antimicrobial activities of essential oils, such as the Juniperus species oil extract [39]. These activities were attributed to their ability to penetrate microbial cells and cause alterations in their structure and function due to their hydrophobic nature. Additionally, their functional diversity and diverse mechanisms of action increased microbial sensitivity [4,14]. Cinnamon oil had high antimicrobial activities, in contrast to other tested plant extracts. Therefore, we recommended the use of cinnamon oil as a food additive to replace other chemical preservatives due to its relatively lower toxicity and side effects [11].

For treating infections with resistant pathogens, a high concentration of cinnamon oil is required [4]; therefore, there is a limitation in its medical uses. This forced us to test the synergistic effect of antibiotics with the best choice of plant extract (cinnamon oil). Our results proved that (i) the usage of this combination overcame the drug resistance; (ii) it decreased the required doses, reducing both adverse/toxic side effects and cost, and (iii) increased the spectrum of activity [4,40].

Without a doubt, the antimicrobial activity of the essential oils is dependent on their chemical composition, which can vary due to many factors which affect the plant’s environment (e.g., geographical location, soil type, weather conditions, etc.) [41]. Therefore, it is important to determine the chemical composition of the essential oils for a correlation with their antimicrobial activities. The chemical profiling of cinnamon oil was performed using GC/MS. The analysis of cinnamon essential oil indicated the presence of four active antimicrobial components: linalool, camphor, (Z)-3-Phenylacrylaldehyde, and 3-phenyl- 2-Propenal. In another study, the gas chromatography and mass spectrometry of the essential oil of Cinnamomum Verum showed that trans-cinnamaldehyde, benzyl alcohol, and eugenol were the major components, which reflected their antimicrobial activities [42].

In another context, the anti-virulence activities of the essential oils were discussed in several studies, and were considered as next generation therapies [43,44]. The computational molecular docking in this study confirms the anti-virulence activities of the pure components of cinnamon oil. The phenylacrylaldehyde and its stereoisomer 2-Propenal,3-phenyl showed the highest binding capacity with the salmonella invasion protein D (SIP-D), which was essential for the internalization of salmonella through modulating the secretion of SipA, SipB, and SipC [45].

## 4. Materials and Methods

### 4.1. Sampling

A total of 100 random samples of meat, chicken, and their products were collected from different markets and butcher shops with different sanitation levels in El-Sharkia and Port Said governorate.

### 4.2. Isolation and Identification

The samples were prepared and *Salmonella* was identified according to the standard bacteriological methods and serotyping techniques, including both tube and slide agglutination techniques [46], and genotypically identified depending on the PCR analysis of 16SrRNA [47]. *Salmonella* isolates were serotyped according to the manufacturer’s instructions using commercial antisera (Difco, Detroit, MI, USA) in the Serology Unit, AHRI, Dokki, Giza, Egypt.

### 4.3. Extraction of Herbals

#### 4.3.1. Oily Extraction

The essential oils were extracted by hydrodistillation in a Clevenger-type apparatus [48]. The essential oils were stored at 4 °C in the dark with the presence of anhydrous sodium sulfate. The essential oils were dissolved in 0.5% DMSO with Tween 80.

#### 4.3.2. Aqueous Extraction

The crude extract was prepared using the method illustrated by [49]. The removal of the extra debris and mud was carried out by washing the fresh plant under tap water 2–3 times. The fine pieces was obtained from the fresh plant and again washed with the distilled water. Pestle and mortar were used to crush these fine pieces to form a fine paste. Whatman filter paper no. 1 was used in the filtration process; the filtered solution was centrifuged at 10,000 rpm for 10 min. The antimicrobial activity of supernatant was evaluated.

#### 4.3.3. Alcoholic Extraction

The 70% methanol was used in the extraction of the dried herbal (20 g) followed by 3 filtration processes. A rotary evaporator was used to concentrate the filtrate at 45 °C for methanol evaporation according to [50] with little modification.

### 4.4. Agar Well Diffusion Assay for the evaluation of Antibacterial Activities

A loopful of bacterial isolates was inoculated into nutrient broth then incubated at 37 °C for 18 h. The bacterial suspensions were normalized according to the standard tube (McFarland number 0.5). A cotton swab was dipped and streaked onto the surface of Mueller-Hinton agar plates, and the plates were left for 5–15 min to dry at room temperature [51]. In the case of the oily form, the oily extract was solubilized with an equal volume of 5% dimethylsulphoxide (DMSO). Then, wells (5 mm diameter) were cut by the cork borer, and then 20 µ of each extract was added. All plates of the tested pathogens were incubated. The zones of inhibitions were measured after 24 h. for all isolates in millimeters (mm).

### 4.5. Antibiotic Sensitivity Testing

#### 4.5.1. Disc Diffusion Method

The antibacterial sensitivity test of the isolates was carried out by Kirby test [52]; each test isolate was swabbed onto the surface of Mueller-Hinton agar plates. Antimicrobial disks including streptomycin (S: 10), sulfamethoxazole-trimethoprim (SXT: 25), gentamicin (CN: 10), cefotaxime (CTX: 30), chloramphenicol (C: 30), amoxicillin-clavulanic acid (AMC: 20/10), doxycycline (DO: 30), ciprofloxacin (CIP: 5), rifamycin (RF: 15), and amikacin (AK: 30) were then placed onto the surface plate of agar, and then incubated. The zones of inhibition were determined in millimeters [53].

#### 4.5.2. Antibiotic Stock Solution Prepared for Effective Antibiotics 

Standard powder forms of effective antibiotics were stored at 4 °C until use. The stock solution of each antimicrobial was adjusted by solubilizing suitable amounts of the antimicrobials in Mueller-Hinton broth to reach a concentration of 1024 μg/mL.

#### 4.5.3. Determination of the Minimum Inhibitory Concentration (MIC) and the Minimum Bactericidal Concentration (MBC)

Microdilution method could be used to determine the MIC values of antibiotics according to [54]. The stock solution of antimicrobial agents (1024 μg/mL) was serially diluted in 96-well microtiter plates containing Mueller-Hinton broth (MHB, Oxoid, Basingstoke, UK) medium. Each well was inoculated with an equal volume of overnight culture of *Salmonella* (5 × 10^5^ CFU/mL) then each plate was incubated. The MIC values were defined as the lowest concentration of antimicrobial agents that completely inhibited microbial growth. Meanwhile, the lowest concentration revealed no visible growth after sub-culturing on the fresh medium was defined as MBC [55].

### 4.6. Measuring the Synergetic Effect of a Herbal–Antibiotic Combination

The best essential oils and plant extracts, combined with antibiotics, were used to study their antibacterial effect on isolated microorganisms; these results were compared with the same essential oils, plant extracts, and antibiotics when used as solo treatments [56].

### 4.7. Evaluation of Active Principle for Effective Extract

#### 4.7.1. Thin Layer Chromatography and Bio-Autography

Active components of cinnamon essential oil (that showed the best result as an antibacterial agent against *Salmonella*) were characterized by thin-layer chromatography (TLC). Silica gelGF254 plates (Merck KGaA, Darmstadt, Germany) were used in a system of toluene:ethyl acetate at a ratio of 93:7 *v/v* in a pre-saturated glass chamber according to [16]. The stationary phase was TLC paper coated with silica gel, while the mobile phase was a solvent system. The spots on the plate were inspected after they were air-dried under UV light (254 nm) and also visualized by spraying with p-anisaldehyde–sulphuric acid reagent then heating for 5 min at 110 °C. The retardation factor value (Rf value) of different separated compounds was calculated by using the formula (Rf = distance traveled by sample/distance traveled by the solvent).

Then, the active constituents were removed from the scrapped silica gel with dichloromethane. The silica gel was removed by centrifugation (12,000× *g*, 15 min). The 0.22 μm filter was used to filtrate the supernatant. Then, 0.1 mL inoculum of *Salmonella* (5 × 105 CFU/mL) was added for every 10 mL of melted nutrient agar. The 100 μL from the supernatant was added to the well, then the plate was incubated. The zones of Inhibitions were compared with the Rf of the spots on the reference TLC plate.

#### 4.7.2. Characterization of the Isolated Compound Using GC-MS Analysis

GC-MS analyses were developed using Shimadzu Japan gas chromatography QP2010PLUS, and the following conditions were applied: temperature programming from 80–2000 °C at 80 °C for 1 min; rate 5 °C/min at 200 °C for 20 min; field ionization detector (FID) temperature 300 °C; injection temperature 220 °C; carrier gas nitrogen at a flow rate of 1 mL/min, and split ratio 1:75. Gas chromatography mass spectrum was performed using GCMS –QP 2010 PlusShimadzu Japan with injector temperature of 220 °C and carrier gas pressure of 116.9 kpa. The elutes were transported into a mass spectrometer. The mass spectrum was programmed with a computer mass-spectra data bank. The computer Wiley MS libraries was used to detect the chemical constituents of the extracts, which were confirmed by comparing mass spectra of the peaks [57,58,59,60].

### 4.8. Molecular Docking on Salmonella Virulence Factor

The crystal structure of the Salmonella Type III Secretion System Tip Protein SipD in complex with deoxycholate (PDB ID: 3O01, RMSD = 1.9 Å) was used for the molecular docking studies. Molecular docking of identified compounds with SipD proteins was performed using ICM-PRO v. 3.9-2b software (MolSoft L.L.C., www.molsoft.com, accessed on 29 October 2021) [61]. ICM-PRO software was also used for the preparation of the structures of molecules and protein, visualization of docked complexes and 2D interaction plots. Standard parameters, recommended by the developers, were used for the execution of molecular docking runs. Docking effort was set to 10. Regular “Score” of ICM-PRO software, which is a GBSA/MM type scoring function augmented with a directional hydrogen bonding term, was used for evaluation of the interaction energy of the studied compounds and protein [62].

### 4.9. Bioinformatics and Statistical Analysis

Statistical Package for Social Sciences software (SPSS; v. 25, IBM, Armonk, NY, USA) was used for statistical analysis of data. Hierarchical clustering (dendrogram) was used to show the overall effect of each treatment on the two serotypes of *Salmonella*. The *Heatmap* package in R was carried out for the inhibition zones (mm) data for each treatment. Regarding the association and overlap among various treatments and their effect on *Salmonella* serotypes, a non-metric multidimensional scaling (nMDS) was generated and visualized using the Euclidean distances among the points as the average of the three replicates and as replicates for each treatment. This analysis was conducted using PC-ORD Software v. 5 (MjM Software, Northampton, UK) [63].

## 5. Conclusions

The current findings indicate that the aqueous extract of paprika and oil extract from cinnamon showed effective antimicrobial activities. For that, *Salmonella* sp. as foodborne pathogens could potentially be managed using new therapeutic agents from natural sources. Therefore, we approved the use of cinnamon oil instead of chemical preservatives, in addition to its therapeutic use when co-administrated with other antibiotics. Hence, further studies must be undertaken to find more promising alternatives and complementary therapies from natural sources.

## Figures and Tables

**Figure 1 antibiotics-10-01453-f001:**
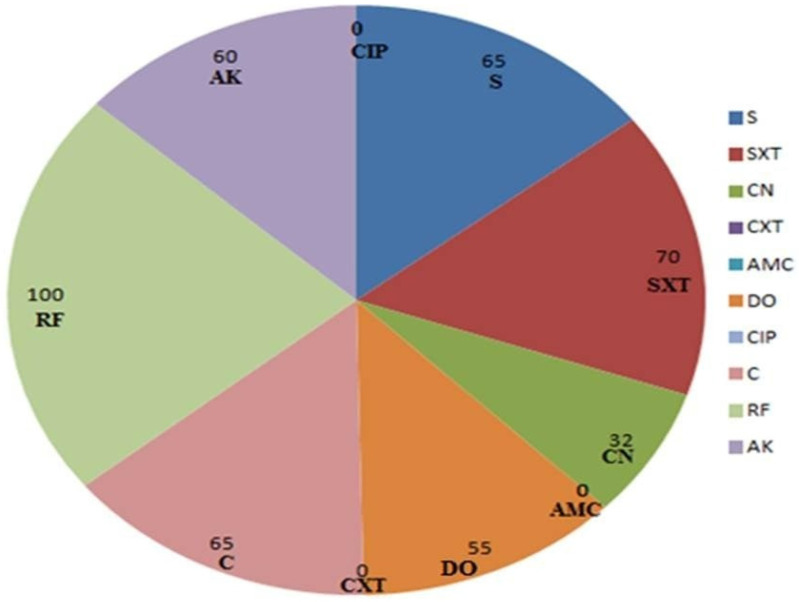
Percentage of the resistant *Salmonella* isolates to each antimicrobial agent. In contrast to rifampicin, the cefotaxime, ciprofloxacin, and amoxicillin+ clavulanic acid showed a maximum antimicrobial activity against all *Salmonella* isolates.

**Figure 2 antibiotics-10-01453-f002:**
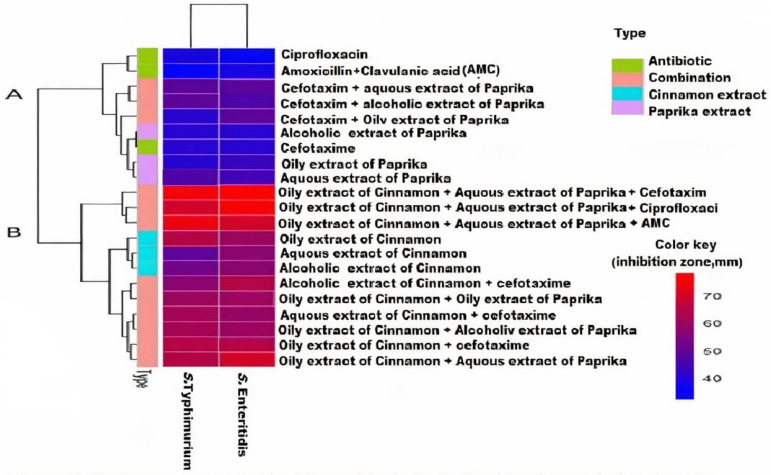
Heat map supported by hierarchical clustering (dendrogram). This figure shows the effect of different treatments against the two species of *Salmonella*. The color key indicates the inhibition zone measured as mm. The analyses were conducted using R program (packagepheatmap). The type refers to the category of the treatment.

**Figure 3 antibiotics-10-01453-f003:**
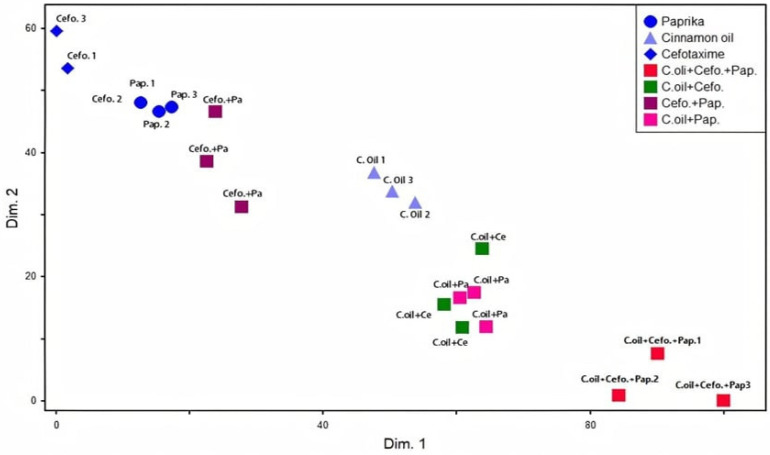
Non-metric multidimensional scaling. The overlap among various treatments against the two *Salmonella* serotypes was visualized. Each dot refers to treatment of one biological replicate. This analysis was conducted using the PC-ORD software. Cefo: cefotaxime, pa: paprika, pa1: aqueous extract of paprika, pa2: alcoholic extract of paprika, pa3: oily extract of paprika, C.oil: cinnamon oil.

**Figure 4 antibiotics-10-01453-f004:**
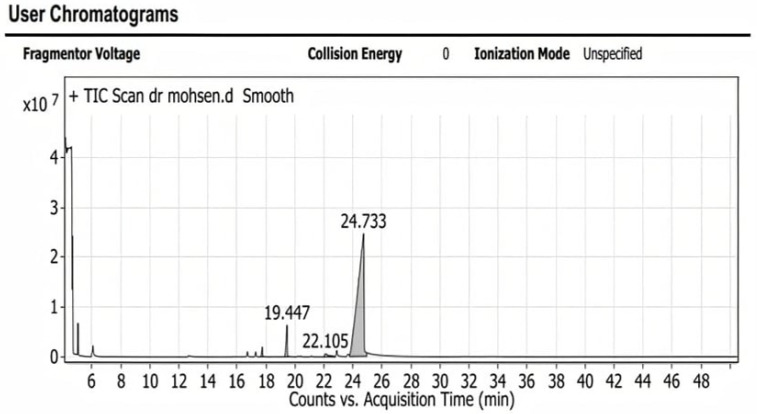
TLC-bioautography of cinnamon oil extract.

**Figure 5 antibiotics-10-01453-f005:**
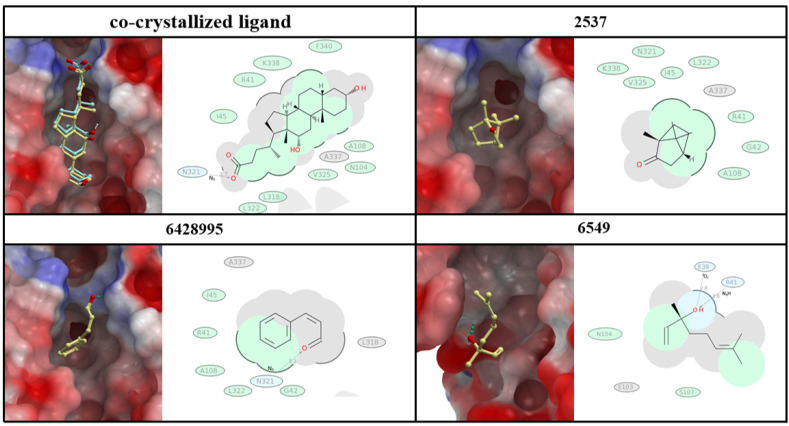
Docking poses and 2D interaction diagrams of selected compounds for *Salmonella*’s SipD protein (PDB ID 3O01). Crystal conformation of co-crystallized ligand is colored in yellow and docked conformation is colored in blue. Green shading in 2D interaction diagrams represents hydrophobic interactions, grey areas, broken thick lines around ligand shape indicates accessible surface, and arrows represent hydrogen bonds.

**Table 1 antibiotics-10-01453-t001:** Molecular docking scores and basic chemical properties of selected and studied ligands.

PubChem CID	Score	Molecular Weight	logP	Heavy Atoms	HB Donor	HB Acceptor	Rotatable Bonds
Co-crystallized ligand (deoxycholate)	−17.02	392.571	4.51	28	3	4	4
Z-3-Phenylacrylaldehyde and its stereoisomer, 2-Propenal,3-phenyl	−14.16	132.16	1.91	10	0	1	2
Camphor	−11.15	152.23	2.4	11	0	1	0
Linalool	−13.19	154.25	2.67	11	1	1	4

## Data Availability

All data generated or analyzed during this study are included in the published article or as Appendix A.

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
