# Peer review of "Alternative and Complementary Therapies against Foodborne Salmonella Infections"

_antibiotics, 2021, doi:10.3390/antibiotics10121453_

Round 1
Reviewer 1 Report
- What are the unrivaled advantages of the form of oil when using a natural compound as a preservation material, and what are the differences from other forms?
- What is the effect of using antibiotics with high concentrations of plant extracts on meat as food?
- What economic advantages does the use of preservatives using natural compounds have over existing synthetic chemicals?
- What should be noted when Natural compound is compared to synthetic chemicals to maintain a constant antibacterial capacity against foodborne pathogens?
Author Response
Reviewer Comments:
Thank you in advance for the precise reviewing. We appreciate the time and detailed feedback provided and we have incorporated the suggested changes into the manuscript to the best of our ability.
1-What are the unrivaled advantages of the form of oil when using a natural compound as a preservation material, and what are the differences from other forms?
Author response: There is a high therapeutic benefit of the essential oils and other plant extracts when they used as a natural preservatives. Essentials oils showed higher inhibitory activity against foodborne pathogens and have a high stability in different pH, temperature and radiations. Additionally, the chemical preservatives are the main causes of hypersensitivity reactions when used in food and pharmaceutical preparation even if they added in small concentration (Wong etal., 2000). Increasingly, there is an urgent need for chemical preservatives free food and pharmaceutical preparation especially for the allergic individuals. Therefore, the essential oils can successfully be used instead of other chemical preservatives.
This was incorporated in our manuscript line (line 64:70)
2-What is the effect of using antibiotics with high concentrations of plant extracts on meat as food?
Author response: The high concentration of plant extracts and essential oils may cause negative organoleptic characteristics in meat, i.e., may alter texture, color, odor, and taste of food. The imparting of the color to meet is the major challenge to the using of these types of natural preservatives. We can overcome this problem by combination therapies that proved its potency in recent years. We can overcome this problem especially the organoleptic impact by combination therapies which allowing the incorporation of essential oils in wide range of food products. Additionally, new nanoencapsulated formulations may aid the wider applications of these compounds (Hyldgaard etal.,2012).
This was incorporated in our manuscript line (line 78:84)
3-What economic advantages does the use of preservatives using natural compounds have over existing synthetic chemicals?
Author response: The cost/benefits ratios of the using of natural preservatives are considered in food industries. The high cost of natural preservatives can limit their used comparing to the synthetic preservatives. The increasing in the economic cost is the main objection for the completely replacement of synthetic preservative by natural one. Thus, the combination between different classes of natural preservatives or with other chemical antibiotics and the using of nanotechnology can reduce this issue (Sojic etal., 2019).
This was incorporated in our manuscript line (line 84:90)
4-What should be noted when Natural compound is compared to synthetic chemicals to maintain a constant antibacterial capacity against foodborne pathogens?
Author response: The extraction methods are the critical point in the maintenance of the bioactive constituents from plant materials. Therefore, the antimicrobial activities of natural compounds are greatly affected by the extraction methods. Of note, there is no universal extraction methods that provide maximum antimicrobial activates for the natural compounds. For each plant and target compounds, there is a unique extraction method. Essentially, the selection of the extraction method is very important and must be evaluated. (Azwanida, 2015).
This was incorporated in our manuscript line (line 91:97 and 99:100)
Reviewer 2 Report
Dear Author(s)
Your paper is interesting, however it needs some improvements as mentioned below in details. all revisions have been also carried out in the attached pdf file.
Line 17: Check the grammer, should be (failure treatment).
Line 19: this is most generalized, you mean specific the studied resistant isolate.
Lines 21,22: This phrase need to be repeated , it is not so clear.
Lines 27,28: Did you analyzed biologically individually these substances , or you just hypothesized this correlation.
Line 41: should be deleted or integrated with the previous phrase.
Line 44: "forced us" in the introduction is not advisable term. please change.
Lines 46-48: Please cite here the following references:
- Elshafie H.S., et al., 2017. Biological activity and chemical identification of ornithine lipid produced by Burkholderia gladioli pv. agaricicola ICMP 11096 using LC-MS and NMR analyses. J. Biol. Res. 90 (6534), 96-103. DOI: 10.4081/jbr.2017.6534.
- Camele I., et al 2019. Bacillus mojavensis: Biofilm formation and biochemical investigation of its bioactive metabolites. J. Biol. Res. 92 (8296), 39-45. DOI: 10.4081/jbr.2019.8296.
- Camele, I., et al., 2019. Anti-quorum Sensing and Antimicrobial Effect of Mediterranean Plant Essential Oils Against Phytopathogenic Bacteria. Frontiers in Microbiology 10, 2619. DOI: 10.3389/fmicb.2019.02695.
Lines 86-89: There is different font here, please correct this.
Figure 1: Please change this figure or add only simple table, it is not clear this figure and the unit of (Y) axis.
Lines 106-110: Correct the font.
Figure 3: Under the figure you should add also the key of meaning symbols.
Line 120: TLC: TLC is not very accurate method, it is enough that you used GC-MS. Or you should declare the differences between two methods.
Line 132: “oily extract “ Did you mean cinnamon oil ?.
Figure 5: It is not advisable to add these structures because they are well known, just mention them in the text.
Lines 177,178: All scientific names use abbreviations, only the first mention use the full name.
Lines 201-203: Please you should also refer to the hydrophobic nature of some single constituents of EOs which enable them to interact with microbial membranes and cause cell lysis, and inhibiting protein synthesis.
as reported by :
- Gruľová, D., et al 2020. Thymol Chemotype Origanum vulgare L. Essential Oil as a Potential Selective Bio-Based Herbicide on Monocot Plant Species. Molecules 25 (3), 595. DOI: 10.3390/molecules25030595.
- Sofo A., et al 2018. Impact of airborne zinc pollution on the antimicrobial activity of olive oil and the microbial metabolic profiles of Zn-contaminated soils in an Italian olive orchard. J. Trace Elem. Med. Biol. 49, 276–284. DOI: 10.1016/j.jtemb.2018.02.01
- Elshafie H.S., et al 2020. Biological investigations of essential oils extracted from three Juniperus species and evaluation of their antimicrobial, antioxidant and cytotoxic activities. J. Appl. Microbiol. 129, 1261—1271.DOI: 10.1111/jam.14723.
Line 247: Correct the symbol (4 °C).
Line 249: should be Italic (v/v).
Paragraph 4.3.2 Change the font.
Line 269: use the same term for hrs ,( h, hrs , hourse).
Line 364: why you used different programs, i think SPSS is enough.
Conclusion: The conclusion should be rewritten again it is very short and did not include an overview about the obtained results.
References: Again the font of references is not as the text.

Author Response
Thank you in advance for the precise reviewing. We appreciate the time and detailed feedback provided and we have incorporated the suggested changes into the manuscript to the best of our ability.
Your paper is interesting, however it needs some improvements as mentioned below in details. all revisions have been also carried out in the attached pdf file.
Author response: Thank you for your positive comment
Line 17: Check the grammer, should be (failure treatment).
Author response: Thank you for your comment. It was rephrased
Line 19: this is most generalized, you mean specific the studied resistant isolate.
Author response: We specify our isolates by adding these isolates which refer to salmonella in the previous sentence
Lines 21,22: This phrase need to be repeated , it is not so clear.
Author response: Thank you for your comment, we agree with your suggestions: It was rephrase
Lines 27,28: Did you analyzed biologically individually these substances , or you just hypothesized this correlation.
Author response: These substances were the major constituent of the cinnamon oil according to GC-MS results and the insilico assay approved their activities
Line 41: should be deleted or integrated with the previous phrase.
Author response: It was done according to your recommendation
Line 44: "forced us" in the introduction is not advisable term. please change.
Author response: It was changed according to your recommendations
Lines 46-48: Please cite here the following references
- Elshafie H.S., et al., 2017. Biological activity and chemical identification of ornithine lipid produced by Burkholderia gladioli pv. agaricicola ICMP 11096 using LC-MS and NMR analyses. J. Biol. Res. 90 (6534), 96-103. DOI: 10.4081/jbr.2017.6534.
- Camele I., et al 2019. Bacillus mojavensis: Biofilm formation and biochemical investigation of its bioactive metabolites. J. Biol. Res. 92 (8296), 39-45. DOI: 10.4081/jbr.2019.8296.
- Camele, I., et al., 2019. Anti-quorum Sensing and Antimicrobial Effect of Mediterranean Plant Essential Oils Against Phytopathogenic Bacteria. Frontiers in Microbiology 10, 2619. DOI: 10.3389/fmicb.2019.02695.
Author response: It was cited
Lines 86-89: There is different font here, please correct this.
Author response: It was corrected
Figure 1: Please change this figure or add only simple table, it is not clear this figure and the unit of (Y) axis.
Author response: The formatting and style in addition to the figure ligand were changed
Lines 106-110: Correct the font.
Author response: It was corrected
Figure 3: Under the figure you should add also the key of meaning symbols.
Author response: It was added
Line 120: TLC: TLC is not very accurate method, it is enough that you used GC-MS. Or you should declare the differences between two methods.
Author response: We totally agree with your comment but we used TLC as a preliminary method as it is easily and cheap method
Line 132: “oily extract “ Did you mean cinnamon oil ?.
Author response: Yes we added this
Figure 5: It is not advisable to add these structures because they are well known, just mention them in the text.
Author response: The figure was removed according your recommendation
Lines 177,178: All scientific names use abbreviations, only the first mention use the full name.
Author response: Thank you for this note; meanwhile, these scientific names were used once in this manuscript
Lines 201-203: Please you should also refer to the hydrophobic nature of some single constituents of EOs which enable them to interact with microbial membranes and cause cell lysis, and inhibiting protein synthesis.
as reported by :
- Gruľová, D., et al 2020. Thymol Chemotype Origanum vulgare L. Essential Oil as a Potential Selective Bio-Based Herbicide on Monocot Plant Species. Molecules 25 (3), 595. DOI: 10.3390/molecules25030595.
- Sofo A., et al 2018. Impact of airborne zinc pollution on the antimicrobial activity of olive oil and the microbial metabolic profiles of Zn-contaminated soils in an Italian olive orchard. J. Trace Elem. Med. Biol. 49, 276–284. DOI: 10.1016/j.jtemb.2018.02.01
- Elshafie H.S., et al 2020. Biological investigations of essential oils extracted from three Juniperus species and evaluation of their antimicrobial, antioxidant and cytotoxic activities. J. Appl. Microbiol. 129, 1261—1271.DOI: 10.1111/jam.14723.
Author response: It was illustrated and referred in our manuscript
Line 247: Correct the symbol (4 °C).
Author response: It was corrected
Line 249: should be Italic (v/v).
Author response: It was corrected
Paragraph 4.3.2 Change the font.
Author response: It was corrected
Line 269: use the same term for hrs ,( h, hrs , hourse).
Author response: It was adjusted
Line 364: why you used different programs, i think SPSS is enough.
Author response: The R-program was used to constract the heat map as in figure 3 and 4
Conclusion: The conclusion should be rewritten again it is very short and did not include an overview about the obtained results.
Author response: it was rewritten again
References: Again the font of references is not as the text.
Author response: It was done
Round 2
Reviewer 1 Report
accept